# LATENT-SPACE SEMI-SUPERVISED TIME SERIES DATA CLUSTERING

## ABSTRACT

Time series data is abundantly available in the real world, but there is a distinct lack of large, labeled datasets available for many types of learning tasks. Semi-supervised models, which can leverage small amounts of expert-labeled data along with a larger unlabeled dataset, have been shown to improve performance over unsupervised learning models. Existing semi-supervised time series clustering algorithms suffer from lack of scalability as they are limited to perform learning operations within the original data space. We propose an autoencoder-based semi-supervised learning model along with multiple semi-supervised objective functions which can be used to improve the quality of the autoencoder's learned latent space via the addition of a small number of labeled examples. Experiments on a variety of datasets show that our methods can usually improve $k$-Means clustering performance. Our methods achieve a maximum average ARI of 0.897, a 140% increase over an unsupervised CAE model. Our methods also achieve a maximum improvement of 44% over a semi-supervised model.

## 1 INTRODUCTION

Time series data can be defined as any data which contains multiple sequentially ordered measurements. Real world examples of time series data are abundant throughout many domains, including finance, weather, and medicine. One common learning task is to partition a set of time series into clusters. This unsupervised learning task can be used to learn more about the underlying structure of a dataset, without the need for a supervised learning objective or ground-truth labels. Clustering time series data is a challenging problem because time series data may be high-dimensional, and is not always segmented cleanly, leading to issues with alignment and noise.

The most basic methods for time series clustering apply general clustering algorithms to raw time series data. Familiar clustering algorithms like hierarchical clustering or $k$-Means clustering algorithms may be applied using Euclidean Distance (ED) for comparisons. Although ED can perform well in some cases, it is susceptible to noise and temporal shifting. The improved Dynamic Time Warping (DTW) (Berndt & Clifford, 1994) metric provides invariance to temporal shifts, but is expensive to compute for clustering tasks. A more scalable alternative to DTW exists in $k$-Shape, a measure based on the shape-based distance (SBD) metric for comparing whole time series (Paparrizos & Gravano, 2017). Shapelet-based approaches such as Unsupervised Shapelets (Zakaria et al., 2012) can mitigate issues with shifting and noise but are limited to extracting a single pattern/feature from each time series.

One alternative approach for clustering time series data is to apply dimensionality reduction through the use of an autoencoder. Autoencoders are capable of learning low-dimensional projections of high-dimensional data. Both LSTM and convolutional autoencoders have been shown to be successful at learning latent representations of time series data. These models can extract a large number of features at each time step. After training an autoencoder model, the learned low-dimensional latent representation can then be fed to an arbitrary clustering algorithm to perform the clustering task. Because autoencoder models reduce the dimensionality of the data, they naturally avoid issues with noise, and provide a level of invariance against temporal shifting.

Recently, the field of semi-supervised learning has shown great success at boosting the performance of unsupervised models using small amounts of labeled data. Dau et al. (2016) proposes a solution for semi-supervised clustering using DTW. However, this solution is still based on DTW, and as

such suffers from scalability issues. He et al. (2019) proposes a constraint-propagation approach for semi-supervised clustering, which may (but is not required to) be used in conjunction with DTW. However, this solution still performs time series comparisons within the raw data space, which may cause issues with scalability for large datasets.

In this paper, we present a semi-supervised deep learning model based on a convolutional autoencoder (CAE), which may be used to perform clustering on time series datasets. We also present new semi-supervised learning objectives, adapted from well-known internal clustering metrics, which can significantly improve clustering performance when provided with a small number of labeled time series. We perform experiments to show that our semi-supervised model can improve performance relative to an unsupervised model when applied for clustering tasks. We also implement a lightly modified batch-based version of the semi-supervised learning solution shown presented in Ren et al. (2018), and show that our proposed solutions are competitive. In the best case, our model semi-supervised model shows a best-case improvement in ARI of 140% over an unsupervised CAE model when applying $k$-Means clustering, and a best-case improvement of 44% over a similar model.

In the remainder of this paper, Section 2 reviews the related work on time series clustering, Section 3 presents our proposed method for semi-supervised time series clustering, and in Section 4.1 we discuss our experimental methodology and present our experimental results. Finally, Section 5 details our conclusions and avenues for future research.

## 2 RELATED WORK

One of the most common ways to perform time series clustering is to apply the $k$-Means algorithm. By default, $k$-Means uses Euclidean Distance (ED). ED is efficient to calculate, and in many cases shows good results (Ding et al., 2008). However, ED comparison will fail when two similar time series are shifted temporally relative to one another. Additionally, ED comparisons are sensitive to noisy data. The Dynamic Time Warping (DTW) metric (Berndt & Clifford, 1994) improves on ED by computing a warping path between a pair of time series. This approach solves issues with temporal shifting, but requires $O(N^2)$ time to compute for two time series of length $N$. Recent work has provided bounds for this computation (Keogh & Ratanamahatana, 2005), (Lemire, 2009), but the scalability of DTW remains an issue for large datasets and long time series. The $k$-Shape algorithm (Paparrizos & Gravano, 2015) is a scalable and performant alternative to DTW, and offers similar performance to DTW at a lower computational cost. The Unsupervised Shapelets (Zakaria et al., 2012) clustering method operates by forming clusters around common subsequnces extracted from the data. This approach provides invariance against shifts since the shapelet may appear anywhere within each time series, and also provides some invariance against noise or outliers within the data, since elementwise comparisons only occur between shapelets, rather than the full time series. In this regard, the UShapelet algorithm has some advantages over DTW and $k$-Shape. However, this method is constrained to extracting a single shapelet/feature from each time series.

Recently, semi-supervised learning has shown the benefit of augmenting a large unlabeled dataset with a small amount of labeled data. There is some existing work for applying semi-supervised learning to time series clustering. The semi-supervised time series clustering solution presented in Dau et al. (2016) proposes a modified version of DTW, which operates in a semi-supervised manner using supervised constraints. However, this method still relies on performing DTW comparison within the original data space, and as such is not a scalable solution for large datasets or long time series. Another methodology for semi-supervised time series clustering is He et al. (2019) which is a graph-based approach using supervised examples to generate positive and negative constraints between points. This approach does not rely on DTW, but the algorithm still performs comparisons in the original data space, which can be problematic as the length of the time series grows.

Popular deep learning frameworks such as LSTMs and CNNs may also be applied to time series data. Both LSTM and CNN networks may be arranged as autoencoders, allowing for unsupervised feature learning for clustering, compression, or anomaly detection tasks. Holden et al. (2015) use a Convolutional Autoencoder (CAE) model to learn a featurized representation of gait data. Autoencoder architectures may also be applied for anomaly detection, as is shown in Bao et al. (2017), where the authors use autoencoders for anomaly detection. Performing comparisons on embedded samples avoids many of the issues of direct pairwise comparisons. Since autoencoders reduce di-

mensionality of the data, distance comparisons will be performed in a lower-dimensional space than the original data, which improves scalability. Embedding the data before comparison also reduces sensitivity to noise, since raw data are not compared directly. One work that takes advantage of this approach is McConville et al. (2019), which learns a manifold for clustering on the trained latent space of an autoencoder. Recent work in the generative model field has produced the Variational Autoencoder (VAE) architecture (Kingma & Welling, 2013), which learns probability distributions for latent features within the data and may also be applied for clustering as in Dilokthanakul et al. (2016). This architecture has also been successfully applied in Fortuin et al. (2018), which uses a VAE based method to learn feature representations of time series. We base our model on a CAE, as the CAE architecture is simple and well-known, and provides a good platform for evaluating the performance of our proposed objective functions.

## 3 SEMI-SUPERVISED LEARNING OF LATENT SPACE

Our proposed framework is a semi-supervised CAE model, which combines an unsupervised reconstruction loss $L_{rc}$ with a semi-supervised loss $L_{sup}$, as shown in Figure 3.1. We show how an existing semi-supervised learning objective function may be adapted to fit into our model, and also propose two new objective functions based on well-known internal clustering metrics. We focus on optimizing the autoencoder's latent space for distance-based clustering algorithms, and specifically perform our experimentation using the $k$-Means algorithm. The spherical, centroid-based clusters generated by $k$-Means are a good fit for the proposed semi-supervised losses, which encourage each cluster to converge around a single high-density point.

### 3.1 CAE MODEL ARCHITECTURE

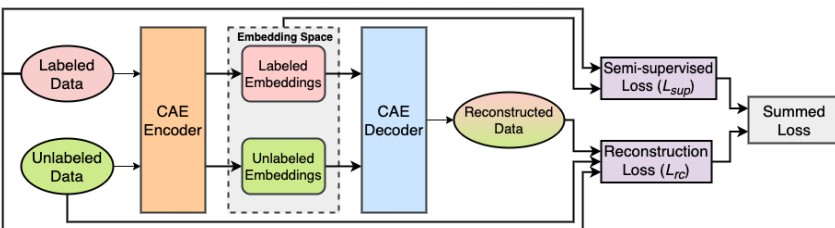

Figure 1: Architecture of Proposed Semi-supervised CAE

We base our model off of a two-layer Convolutional Autoencoder (CAE) architecture. The CAE uses two 1-D convolutional layers to featurize the time series. The filter widths for both layers are calculated at runtime from the length of the time series. Each layer of the encoder contains a 1D convolution operation, followed by a non-linear activation. These layers are paired with transpose convolution layers in the decoder. After each convolution layer in the encoder, we also apply a max-pooling operation to further reduce the length of the featurized sequence. Each max-pooling operation is "reversed" using a nearest-neighbor upsampling operation in the decoder. Alternatively, large strides in the convolutional layer may be used instead of pooling operations. This accomplishes the same goal of reducing the length of the featurized sequence, but does not require any up-sampling operations in the decoder, since the transpose convolution operation with a stride will perform the upsampling. We found that the max-pooling and large stride methods produced similar results in practice.

### 3.2 SEMI-SUPERVISED LOSS FUNCTIONS

#### 3.2.1 PROTOTYPE LOSS

Snell et al. (2017) present a system for performing few-shot learning, where a small number of labeled examples for each class in the dataset are embedded using an encoder network. These points are divided into two sets, query examples and support examples. In the latent space, the support embeddings are averaged to determine a centroid or "prototype" for each class. Training occurs by

measuring the distance from each query example to the support centroid for the class. Each query point is labeled with the class of the closest support prototype. The probability of a query point $i$ belonging to Class $J$ is calculated as:

$$p_{ij} = \frac{e^{-d(i,C_j)}}{\sum_{j'} e^{-d(i,C_{j'})}} \tag{1}$$

In Equation 1, $-d(i,j)$ is the negative distance between query point $i$ and the support prototype for Class $J$. Any distance metric may be used for $d$, but for the remainder of this paper, we treat $d$ as the squared euclidean distance. Training proceeds by minimizing the cross entropy between the labels for the query points and the probability distribution $p$. The objective of this approach is to learn a latent space where similarities between examples are quantifiable using a distance metric like Euclidean distance. Examples from the same class will be close to one another in the embedded space, while examples from separate classes will have larger distances. In addition to the few-shot learning scenario, a later work by Ren et al. (2018) demonstrates that the prototypical network objective can be modified to support semi-supervised few-shot classification as well. The first objective function proposed in Section 3.1.1 of Ren et al. (2018) is calculated in a two-stage process. In the first stage, the labeled data prototypes are calculated as the centroid of the $N_j$ embedded labeled data points $h(x)$ for each class $J$, as shown in Equation 2a. The second stage assigns each unlabeled point a soft cluster weighting, based on the distances to each of the calculated prototypes $C_j$, shown in Equation 2b. Finally, the original prototypes are updated to include the unlabeled data using the soft cluster assignments from Equation 2b. In Equation 2c, the final prototype calculation is performed by adding the embedded values for the labeled class $J$ along with the weighted unlabeled values from set $U$. This sum is then divided by the sum of $N_j$ and $\sum_j w_{i,j}$ which capture the number of labeled examples, and the sum of weights for class $J$, respectively.

$$C_j = \frac{1}{N_j} \sum_{i \in J} h(x_i) \qquad w_{ij} = \frac{e^{-||h(x_i)-C_j||^2}}{\sum_{j'} e^{-||h(x_i)-C_j'||^2}} \qquad \hat{C}_j = \frac{\sum_{i \in J} h(x_i) + \sum_{i \in U} h(x_i)w_{ij}}{N_j + \sum_j w_{ij}} \tag{2a,b,c}$$

We extend our vanilla CAE model into a semi-supervised learning context by calculating refined prototypes from the labeled and unlabeled embeddings within the CAE's embedded space. The semi-supervised loss objective $L_{proto}$ can be written as Equation 3. In Equation 3, $N$ represents the total number of samples in the batch (unlabeled and labeled), $y_{ij}$ is the ground truth indicator for Class $j$, and $\hat{p_{ij}}$ is probability that sample $i$ belongs to class $J$ by performing the calculation in Equation 1 using the refined prototypes from Equation 2c. Figure 1 presents the full architecture of our model.

$$L_{proto} = \frac{1}{N} \sum_i^N \sum_j^K y_{ij} * log(\hat{p_{ij}}) \tag{3}$$

### 3.2.2 SILHOUETTE LOSS

When applying $k$-Means clustering to an unlabeled dataset, one must first choose the correct $k$, or number of clusters to fit using the model. One metric for determining the correct number of clusters is the Silhouette score Rousseeuw (1987). Silhouette score belongs to the family of internal clustering metrics (Maulik & Bandyopadhyay, 2002), which provide a method for evaluating the performance of a clustering algorithm when no ground truth labels are available. In the absence of labels, internal clustering metrics instead evaluate the partitioning's ability to separate data into clusters which have low intra-cluster distance, but high inter-cluster distance. Silhouette is a per-sample metric calculated using the following formulae:

$$a(i) = \frac{1}{|C_k|-1} \sum_{l \in C_k} d(i,l) \qquad b(i) = \min_{k \neq i} \frac{1}{|C_k|} \sum_{l \in C_k} d(i,l) \qquad s(i) = \frac{b(i)-a(i)}{\max a(i),b(i)} \tag{4a,b,c}$$

As mentioned in Section 3.2.1, $d$ is an arbitrary distance metric, but we set $d$ as the squared euclidean distance for all experiments. Equation 4a represents the average intra-cluster distance from point $i$ to all other points with the same cluster label. Equation 4b represents the average distance from $i$ to the second closest cluster, or the inter-cluster distance. The second closest cluster is defined as the cluster having the second lowest average distance. The silhouette score is then calculated as

the difference between the inter-cluster distance and the intra-cluster distance in Equation 4c. To normalize the term, the difference is divided by the maximum of $a(i)$ and $b(i)$. In our configuration, we use silhouette score as a semi-supervised training objective, by providing ground-truth cluster assignments for the labeled points, and calculating cluster assignments for the unlabeled points. In this configuration, Silhouette score will represent the separability of our labeled points within the embedded space. Labeled points will be encouraged to have similar embeddings in the latent space by Equation 4a, and will also be encouraged to separate themselves from embeddings with a different label by Equation 4b. The silhouette values for the labeled points can be generated directly from equations 4a,b,c. To calculate the silhouette values for the unlabeled points, we calculate the closest embedded centroid for each unlabeled point, then calculate Equations 4a and 4b using distances between the unlabeled point and the labeled points in the closest and second-closest clusters respectively. After the silhouette scores are calculated for both the labeled and unlabeled points, we concatenate both groups and take the mean over the batch. We can rewrite Equation 4c as as training objective:

$$L_{silh} = \frac{1}{N} \sum_i^N 1 - s(i) \tag{5}$$

To formulate Equation 5, we take $l(i) = 1 - s(i)$ as a loss term, which has a minimum at $s(i) = 1$ and a maximum at $s(i) = -1$. We take the mean over all $l(i)$ to produce a scalar loss value, representing the separability of our labeled examples within the embedded space. We can train on Equation 5 using gradient descent methods either alone, or in combination with a CAE reconstruction method.

### 3.2.3 DB INDEX LOSS

The Davies-Bouldin index is another example of a internal clustering metric. Similar to the Silhouette score, the DB Index value is a measure of cluster quality, with lower values indicating higher quality clusters. Like the Silhoutte index, the DB Index is comprised of two terms, which are combined to form the metric value.

$$S(C_k) = \frac{1}{N} \sum_{i \in C_k} d(i, \bar{C}_k) \quad M(C_k, C_l) = d(\bar{C}_k, \bar{C}_l) \quad R(C_k, C_l) = \frac{S(C_k) + S(C_l)}{M(C_k, C_l)} \tag{6a,b,c}$$

Equations 6a and 6b capture notions of intra- and inter-cluster similarity, respectively. Equation 6c is a metric which captures the quality of the separation between clusters $C_k$ and $C_l$ as defined by their individual densities, as well as the distances between their centroids $\bar{C}_k$ and $\bar{C}_l$. Lower values of $R$ indicate a higher quality of separation between clusters $C_k$ amd $C_l$, thus $R$ should be minimized. DB Index differs from Silhouette in that the DB Index methods are calculated on each pair of clusters, whereas the Silhouette index is calculated for each sample individually. Equation 7 forms our trainable loss function. As with our implementation for Silhouette loss, we calculate the Equation 7 for both labeled and unlabeled points by assigning unsupervised points a label based on the closest labeled cluster centroid.

$$L_{db} = \sum_{i \neq j} R(C_i, C_j) \tag{7}$$

## 4 PERFORMANCE EVALUATION

In this following section, we evaluate the performance of our method using our unsupervised CAE as the unsupervised baseline result, and the lightly modified Prototype Loss as a comparable semi-supervised result.

### 4.1 EXPERIMENTAL METHODOLOGY

### 4.1.1 MODEL SETUP

For our experimental setup, we use a two-layer CAE model, where convolutional operations are applied along the temporal dimension to featurize the data. In order to perform a fair comparison on multiple datasets, we chose the hyper-parameters for the convolutional layers as follows. For both convolutional layers, we set the filter size as $f = \lfloor \frac{T}{10} \rfloor$, where $T$ represents the length of the

time series. In this way, we determined the filter widths for both layers automatically from the data. For optimization, we use the Adam optimizer (Kingma & Ba, 2014) as implemented in Tensorflow 2.1.0. We use the default learning rate of $lr = 0.001$ for all experiments, and train for 200 epochs. In experiments which apply two loss functions simultaneously, such as the experiments using a semi-supervised loss along with the reconstruction loss, we optimize the sum of the losses. In practice, we found that weighting the losses was not necessary as the model was able to optimize both objectives simultaneously.

### 4.1.2 DESIGN OF EXPERIMENTS

In our experiments, we integrate both the adapted Prototype Loss from Section 3.2.1 and the proposed Silhouette and DB Index Losses from Sections 3.2.2 and 3.2.3 into our CAE architecture as presented in Figure 1 and measure the clustering performance as indicated by the Adjusted Rand Index (ARI) (Hubert & Arabie, 1985), using the scikit-learn implementation of ARI (Pedregosa et al., 2011) for our experiments. In order to perform a fair comparison, we performed tests using the proposed "combined" architectures, which integrate both the semi-supervised losses $L_{sup}$ and the CAE's reconstruction loss $L_{rc}$ and also measured the effect of disabling the CAE's reconstruction loss, and training solely on the proposed semi-supervised architectures. This allows us to isolate the performance improvement for the semi-supervised architectures. Finally, we present a baseline comparison against the CAE's unsupervised performance, using only the reconstruction loss value. The goal of these tests is to demonstrate the performance of each model as the number of supervised examples per-class increases. We perform 4 groups of tests, with each group using a fixed number of supervised examples per-class in the range $[4, 28]$. We initialize 5 training runs for each model within each group. Within a group we use the same 5 random seeds for each model initialization to ensure that the supervised examples chosen, as well as the parameter initializations are all identical within the group. After training, we use the latent space of the trained model to perform $k$-Means clustering, and record the ARI. We use the $k$-Means algorithm because the centroid-based nature of $k$-Means is a natural fit for the proposed losses. Notably, the Prototype Loss corresponds almost exactly to a $k$-Means objective, and both Silhouette and DB Index loss also rely on notions of cluster density around a centroid. However, any other general clustering method may be applied. Labeled examples are included when fitting $k$-Means, but are not included in the ARI metric, to avoid inflating ARI artificially as the number of labeled examples increases.

### 4.1.3 DATASETS

For our testing, we utilize three datasets chosen from the UCR Archive (Dau et al., 2018). All UCR Archive datasets are labeled, which is useful for our evaluation since we may experiment with differing amounts of labeled data. In a real-world scenario with unlabeled data, domain experts provide label information for a small subset of the data. The three datasets that we chose are some of the largest within the UCR Archive. In the case of trainable architecture like our proposed model, large datasets are advantageous, as larger numbers of samples will increase the quality of the latent featurization, and help to improve generalization of the features for unseen samples. All three datasets contains samples which are of the same length. In general , the CAE architecture requires that all samples be the same length, although datasets with variable-length samples can still be used by first applying interpolation or zero-padding to normalize the samples to a consistent length. *FacesUCR* is a dataset containing face outlines from 14 different individuals, represented as 1D time series. *ECG5000* is a dataset containing ECG readings of heartbeats from a congestive heart-failure patient. *UWaveGestureLibraryAll* is a dataset containing accelerometer recordings of subjects performing different types of hand gestures. Table 2 in Appendix A presents a summary of the characteristics of each dataset.

### 4.2 EXPERIMENTAL RESULTS

The results for the tests on all three datasets are presented in Figure 3. To provide a reference for the performance of our models relative to the unsupervised models, we also present Table 1, which provides ARI performance figures for the $k$-Means and $k$-Shape unsupervised clustering algorithms, as applied to the raw data for each of our chosen datasets.

| Dataset | $k$-Means | | $k$-Means + PCA | | $k$-Shape | |
|---|---|---|---|---|---|---|
| | $\mu$ | $\sigma$ | $\mu$ | $\sigma$ | $\mu$ | $\sigma$ |
| ECG5000 | **0.555** | **0.092** | 0.515 | 0.011 | 0.441 | 0.085 |
| FacesUCR | 0.223 | 0.037 | 0.199 | 0.007 | **0.441** | **0.037** |
| UWaveGestureLibraryAll | **0.632** | **0.055** | 0.612 | 0.045 | 0.238 | 0.027 |

Table 1: Unsupervised methods performance

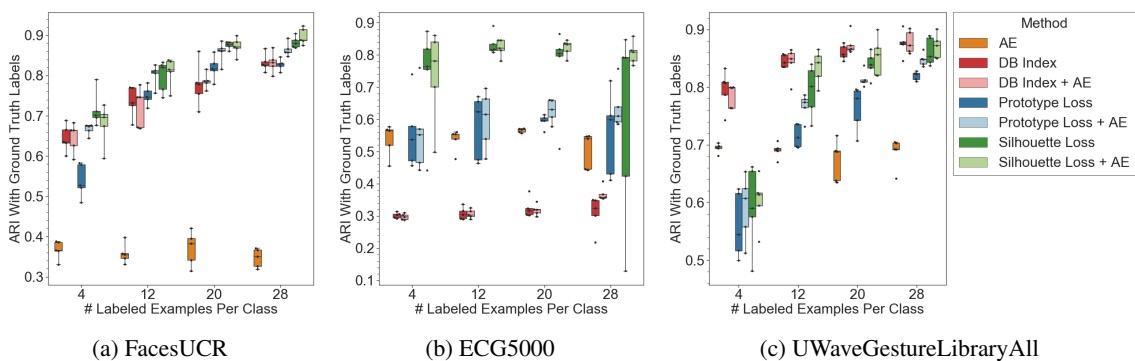

(a) FacesUCR      (b) ECG5000      (c) UWaveGestureLibraryAll

Figure 2: Overall performance results for selected datasets

### 4.2.1 OVERALL PERFORMANCE

The results for *FacesUCR* are presented in Figure 2a. As shown in the figure, the semi-supervised approaches significantly improve the performance of the model, relative to the baseline CAE ARI of 0.35. According to Table 1, the CAE's performance here is much better than the $k$-Means performance on raw data, but slightly worse than $k$-Shape, which achieves an average ARI of 0.441. Out of all the semi-supervised models, the Silhouette Loss + AE model performs slightly better on average than the other models, including the Silhouette Loss without AE. This dataset is an excellent candidate for this type of semi-supervised model, as even providing 4 labels per class can achieve an ARI of 0.7 when using the Silhouette Loss. We can see that as the number of supervised examples per-class increases, the ARI achieved for all of the semi-supervised models also increases, approaching a 0.9 ARI. All semi-supervised models perform well on this dataset, although the two Silhouette Loss models seem to have a slight edge over the others.The results for *ECG5000* in Figure 2b show that both DB Index methods perform poorly on this dataset. Both the DB Index and DB Index + AE methods perform worse than the baseline CAE at all numbers of labeled examples. The baseline CAE model performs similarly to $k$-Means clustering on ECG5000, and better than $k$-Shape. The Prototype and Prototype + AE Methods do show some improvement over the baseline CAE, but these improvements are relatively minor. The Silhouette and Silhouette + AE methods outperform all other methods here, achieving a ARI of 0.8 for all numbers of labeled examples. However, two of the Silhouette Loss trials at 28 examples seem to diverge, and produce poor results. The Silhouette Loss + AE models do not suffer this same divergence, and provide a stable ARI of around 0.8 for all trials at 28 examples per-class. We suspect that the AE model and associated $L_{rc}$ was able to help mediate the effect of the divergence in the Silhouette + AE model. The Silhouette + AE model does encounter one minor divergence at 4 examples per-class, where it only achieves a ARI of 0.5. However, we expect that smaller numbers of labeled examples will tend to be noisier, since the model performance depends heavily on the choice of supervised examples. In this case, the Silhouette models are the winners, but do suffer from some divergence issues as mentioned before. We believe that this issue is caused in part by the extremely unbalanced nature of the ECG5000 dataset. Table 3 in Appendix A shows the distribution of classes. Two of the classes are very sparse, and must be over-sampled during training in order to provide the correct number of labeled examples. Additionally, most of the cluster points are concentrated in Clusters 1 and 2. During training, the clusters with smaller numbers of ground-truth labels tend to "steal" some of the true members of Clusters 1 and 2, leading to a converged result where Clusters 3, 4, and 5 are much

larger than in the ground-truth data. The results for *UWaveGestureLibraryAll* in Figure 2c show that DB index performs consistently well over all numbers of labeled examples. At 4 labeled examples per-class, the DB Index and DB Index + AE methods are the only methods which outperform the standard baseline CAE. Starting at 12 examples per-class, the Silhouette and Prototype models do start to outperform the CAE baseline, but perform noticeably worse than the DB Index models until 28 examples per-class. For this dataset, even the baseline CAE outperforms *k*-Means on the raw data, which performs the best of the three unsupervised models in Table 1. In this experiment, the DB Index models are the clear winner since they provide excellent performance for any number of labeled examples tested.

### 4.2.2 HYPERPARAMETER STUDY

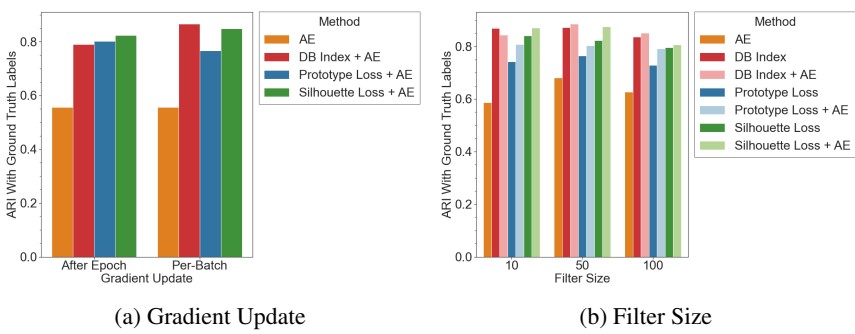

|  (a) Gradient Update | (b) Filter Size |

Figure 3: Hyperparameter Study Results

In the performance evaluation results from Section 4.2.1, we perform parameter updates at the end of each batch. In order to better understand how the frequency of parameter updates affects the overall performance, we also experiment by applying the update for the semi-supervised loss at the end of each epoch, while still updating parameters from the autoencoder loss at the end of each batch. In order to accomplish this, we calculate the semi-supervised gradients at each batch, accumulating them and applying the sum as the gradient update at the end of each epoch. For this experiment, we train the model on the *UWaveGestureLibrary* dataset and choose 12 supervised examples per class. Since autoencoder updates are performed at the end of each batch, the autoencoder result does not change based on gradient update method, and the results are only provided for comparison. We train all versions of the model using the same random seed, only varying the gradient update method between the two model instances. Figure 3a shows the result of the experiment. As expected, the AE model obtains identical performance between the two update methods. The DB Index and Silhouette methods see a marginal improvement when training using the per-batch methodology. The Prototype Loss method sees marginally better performance when updating at the end of the epoch. In Section 4.1, we describe our method for determining the convolutional filter size dynamically based on data input. In this experiment, we test the same model setup as the gradient test, but vary the filter size. Gradient updates are performed at the end of each epoch. Figure 3b shows the result of this experiment. Most models perform their best with the filter size of 50, but in general performance does not differ much with different choices for filter size. In the real-world use case, a sub-optimal choice for convolutional filter size should not degrade the performance of the model.

We also explored the usability of the learned latent space for classification by applying a KNN classification. For this series of tests, we treat the randomly chosen "labeled" examples for each class as the training points for a KNN classifier, then predict the class of the unlabeled points. We calculate the accuracy of the KNN classifier for each of the three datasets described in 4.1.3.

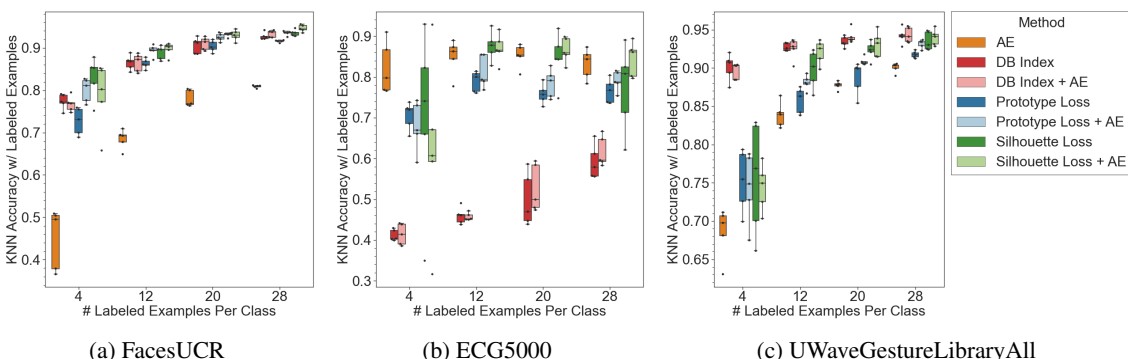

Figure 4: Classification performance for three selected datasets

In the results for *FacesUCR* as seen in Figure 4a, we see a clear distinction in performance between the semi-supervised models and the unsupervised autoencoder. Even for low numbers of supervised examples, all semi-supervised models outperform the autoencoder by a large margin, and the autoencoder never closes the gap in performance, even for larger numbers of supervised examples. The results for *ECG5000* show that all models struggle to perform better than the standard CAE. We suspect that the model performs poorly here because of the imbalanced class size, as mentioned previously. The results for *UWaveGesture* as shown in Figure 4c show more mixed performance. At low numbers of samples, the Sihouette and Prototype losses perform marginally better than the baseline AE, but show a large variance in performance. This demonstrates that both these models' performance is highly sensitive to the choice for labeled examples. The DB Index models perform distinctly better than any other on this dataset, and show little variance in performance even for lower numbers of labeled examples. We suspect that this is because the embedding clusters in *UWaveGesture* are more distinct from each other, so the DB Index approach, which is based on optimizing distances between pairs of clusters, performs the best. The baseline AE also shows a large improvement here as the number of labeled examples increases, although it does not outperform the semi-supervised models.

## 5    CONCLUSION

In this paper, we proposed a framework for semi-supervised time series clustering, including three alternative semi-supervised loss functions. Our experiments show that all three implemented semi-supervised models can improve clustering performance after training. Experiments also show that training the semi-supervised losses in combination with the reconstruction loss from the autoencoder does provide a slight boost in performance, although this difference is usually small. Although all solutions have generally stable performance across multiple parameter initialization and choices for supervised examples, we do see occasional model divergences. Because these models rely on the labeled examples for training, the quality of these labels is exceedingly important. In a real-world usage scenario, we expect a data domain expert providing labels would be able to choose the most relevant examples to label. Additionally, datasets with significantly unbalanced class sizes may cause performance issues, as are exhibited by our models' performance on the *ECG5000* dataset. The results in Tables 4-6 show that Silhouette Loss on average outperforms the baseline CAE, except in the 4-sample case on the *UWave* dataset. In addition, the DB Index loss on average outperforms the CAE on both the *UWave* and *FacesUCR* datasets.

In future work, we plan to explore methods for combining the predictions of these models by training multiple instances of the same model in parallel, then using a consensus clustering system to generate the final set of labels. We expect that this will reduce the severity of model divergences. In a similar vein, we will explore a way to determine an optimal weighting between the reconstruction and semi-supervised losses, since our method currently applies no weighting. Additionally, we believe that training multiple models simultaneously and applying mutli-view learning constraints like those proposed in Wang et al. (2015) could improve the quality of the model's generated latent space.

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

## A  APPENDIX

| Dataset | Samples | Classes | Length |
|---|---|---|---|
| FacesUCR | 2250 | 14 | 131 |
| ECG5000 | 5000 | 5 | 140 |
| UWaveGestureLibraryAll | 4477 | 8 | 315 |

Table 2: Specifications of the selected datasets

| Cluster | N (%) |
|---|---|
| 1 | 2919 (58.38%) |
| 2 | 1767 (35.34%) |
| 3 | 96 (1.92%) |
| 4 | 194 (3.88%) |
| 5 | 24 (0.48%) |

Table 3: Specifications of *ECG5000*

| Examples | Method | ARI | |
| --- | --- | --- | --- |
| | | $\mu$ | $\sigma$ |
| 4 | AE | 0.694 | 0.008 |
| | **DB Index** | **0.796** | **0.034** |
| | DB Index + AE | 0.785 | 0.020 |
| | Prototype Loss | 0.560 | 0.057 |
| | Prototype Loss + AE | 0.591 | 0.056 |
| | Silhouette Loss | 0.592 | 0.073 |
| | Silhouette Loss + AE | 0.602 | 0.045 |
| 12 | AE | 0.690 | 0.013 |
| | **DB Index** | **0.845** | **0.011** |
| | DB Index + AE | 0.842 | 0.027 |
| | Prototype Loss | 0.715 | 0.020 |
| | Prototype Loss + AE | 0.767 | 0.021 |
| | Silhouette Loss | 0.793 | 0.044 |
| | Silhouette Loss + AE | 0.835 | 0.029 |
| 20 | AE | 0.673 | 0.036 |
| | DB Index | 0.860 | 0.013 |
| | **DB Index + AE** | **0.874** | **0.018** |
| | Prototype Loss | 0.764 | 0.038 |
| | Prototype Loss + AE | 0.814 | 0.014 |
| | Silhouette Loss | 0.839 | 0.022 |
| | Silhouette Loss + AE | 0.853 | 0.034 |
| 28 | AE | 0.687 | 0.026 |
| | DB Index | 0.876 | 0.021 |
| | **DB Index + AE** | **0.878** | **0.019** |
| | Prototype Loss | 0.819 | 0.007 |
| | Prototype Loss + AE | 0.847 | 0.011 |
| | Silhouette Loss | 0.861 | 0.024 |
| | Silhouette Loss + AE | 0.871 | 0.021 |

Table 4: UWaveGestureLibrary Result Table

| Examples | Method | ARI | |
| --- | --- | --- | --- |
| | | $\mu$ | $\sigma$ |
| 4 | AE | 0.537 | 0.051 |
| | DB Index | 0.302 | 0.009 |
| | DB Index + AE | 0.297 | 0.009 |
| | Prototype Loss | 0.556 | 0.114 |
| | Prototype Loss + AE | 0.558 | 0.125 |
| | Silhouette Loss | 0.730 | 0.168 |
| | **Silhouette Loss + AE** | **0.736** | **0.147** |
| 12 | AE | 0.537 | 0.034 |
| | DB Index | 0.307 | 0.019 |
| | DB Index + AE | 0.306 | 0.014 |
| | Prototype Loss | 0.577 | 0.100 |
| | Prototype Loss + AE | 0.598 | 0.090 |
| | **Silhouette Loss** | **0.832** | **0.033** |
| | Silhouette Loss + AE | 0.821 | 0.027 |
| 20 | AE | 0.565 | 0.006 |
| | DB Index | 0.324 | 0.030 |
| | DB Index + AE | 0.318 | 0.017 |
| | Prototype Loss | 0.594 | 0.020 |
| | Prototype Loss + AE | 0.626 | 0.035 |
| | Silhouette Loss | 0.758 | 0.142 |
| | **Silhouette Loss + AE** | **0.822** | **0.025** |
| 28 | AE | 0.504 | 0.055 |
| | DB Index | 0.308 | 0.054 |
| | DB Index + AE | 0.368 | 0.022 |
| | Prototype Loss | 0.554 | 0.131 |
| | Prototype Loss + AE | 0.636 | 0.071 |
| | Silhouette Loss | 0.596 | 0.311 |
| | **Silhouette Loss + AE** | **0.806** | **0.035** |

Table 5: ECG5000 Result Table

| Examples | Method | ARI | |
| --- | --- | --- | --- |
| | | $\mu$ | $\sigma$ |
| 4 | AE | 0.366 | 0.023 |
| | DB Index | 0.644 | 0.034 |
| | DB Index + AE | 0.645 | 0.037 |
| | Prototype Loss | 0.539 | 0.042 |
| | Prototype Loss + AE | 0.666 | 0.014 |
| | Silhouette Loss | 0.715 | 0.044 |
| | **Silhouette Loss + AE** | **0.679** | **0.051** |
| 12 | AE | 0.357 | 0.025 |
| | DB Index | 0.735 | 0.038 |
| | DB Index + AE | 0.722 | 0.049 |
| | Prototype Loss | 0.750 | 0.023 |
| | Prototype Loss + AE | 0.802 | 0.027 |
| | Silhouette Loss | 0.798 | 0.040 |
| | **Silhouette Loss + AE** | **0.809** | **0.036** |
| 20 | AE | 0.370 | 0.043 |
| | DB Index | 0.777 | 0.055 |
| | DB Index + AE | 0.786 | 0.019 |
| | Prototype Loss | 0.819 | 0.029 |
| | Prototype Loss + AE | 0.858 | 0.026 |
| | **Silhouette Loss** | **0.875** | **0.009** |
| | Silhouette Loss + AE | 0.872 | 0.022 |
| 28 | AE | 0.346 | 0.023 |
| | DB Index | 0.832 | 0.022 |
| | DB Index + AE | 0.832 | 0.026 |
| | Prototype Loss | 0.827 | 0.014 |
| | Prototype Loss + AE | 0.864 | 0.017 |
| | Silhouette Loss | 0.882 | 0.014 |
| | **Silhouette Loss + AE** | **0.897** | **0.020** |

Table 6: FacesUCR Result Table

