# OpenReview forum: "Latent Space Semi-Supervised Time Series Data Clustering"
_ICLR.cc/2021/Conference — Reject_

### Official Review · AnonReviewer2 · 2020-10-26
**Review - Some nice ideas but has weaknesses**

**Rating:** 4
**Confidence:** 3

**Review:**

Disclaimer: I am not an expert in the time-series domain, although I did some literature review while performing this review.

#####################################
Summary:
The work investigates semi-supervised (SS) clustering of time-series data (i.e., clustering with few labelled points, which can also be seen as semi-supervised classification here). It specifically investigates how to train a convolutional autoencoder (CAE), which is applied on the time-series data creating an embedding for each timeseries, so that CAE’s embedding clusters samples appropriately. It investigates 3 semi-supervised (SS) losses for this purpose: 1 is the loss proposed in (Ren 2018), and 2 novel losses, the “Silhouette loss” and the “DB index” loss, inspired by the corresponding internal clustering metrics. The work evalutes on 3 databases whether these losses improve clustering of the CAE’s embedding, over vanilla training of the CAE (just reconstruction loss). The work also performs a study of whether the size of the convolution filters affect results and whether performing the gradient updates by the SS loss every batch or every epoch makes a difference.

######################################
Reasons for score:
I am recommending a rejection, because of limited literature review, problems with writing/clarity, limited evaluation and analysis of the experimental setup (configuration etc), and the results are not too strong either. I would not mind much about the performance, if the other weaknesses were not there, as at least the Silhouette loss seems reasonable. But overall, the weaknesses overcame the good points.

#########################################

Pros:
+ Exploration of semi-supervision for clustering is interesting as it’s of practical importance, and rather timely, as there is general interest in the community on SS.

+ The work explores 3 different losses for inducing clustering in the embedding space, 2 of which seem novel. The 2 new losses are quite intuitive as they are inspired by well established metrics. They are generic (not Time-series specific) and could potentially be interesting to parts of the community that look into how to create more discriminative models by clustering the embedding space of deep networks better, such as in semi-supervised learning.

+ Performance of Silhouette loss seems to be interesting, as in all three databases it performs ok. (although I am not 100% convinced due to some unclear points about experimental setup and not too extensive evaluation, see below).

#########################################
Cons:

Con.1:
The work is not well positioned within the literature. The review is limited both from the scope of unsupervised clustering, as well as from the point of view of semi-supervised learning (both in general and specifically to time-series). Details on literature review:

One one hand, the work discusses only “shallow” unsupervised clustering methods, but misses deep models for unsupervised clustering. As a result, it considers as the *only* baseline a vanilla Auto-Encoder (AE). However, for unsupervised clustering, AE has been long surpassed by more appropriate models for clustering (AE has by itself no incentive to cluster the latent space). Examples include Contrastive AEs, extensions of VAE and GANs, etc. I below give a citation to a recent work that extends VAEs for clustering [1], which also contains references to other models, which the authors can consult. Also, some old and recent works that apply such models to time-series are [2, 3, 4]. Hopefully references therein will be of interest to authors. I wouldn’t expect all of them to be evaluated, but at least some discussed, to give a good picture of where the field is on time-series clustering. (A suggestion: Perhaps the authors could improve the work by arguing that they chose AE for its simplicity, to focus specifically on the effect of the loss. But at least this discussion should be made.)

[1] Dilokthanakul et al, Deep unsupervised clustering with gaussian mixture variational autoencoders, 2016.

[2] Rifai et al, Contractive Auto-Encoders: Explicit Invariance During Feature Extraction, 2011.

[3] Fortuin et al, SOM-VAE: Interpretable Discrete Representation Learning on Time Series, ICLR 2019

[4] McConville et al, N2D: (Not Too) Deep Clustering via Clustering the Local Manifold of an Autoencoded Embedding, ICPR 2020 (arxiv Aug 2019)

Within the scope of the time-series application, authors mention deep approaches based approaches (via LSTM and CNN) but they don’t cite any. I would expect some important works to have been cited.

From the point of view of semi-supervised learning (SSL) in deep-learning, the literature review is also limited, constrained to Ren et al (2018). This should be improved. I below provide a reference to a work that proposed an SSL loss for improved clustering of latent-space via label propagation, which is in the same spirit as the losses discussed in this paper and hence should be discussed [5]. I also provide a reference to a more recent work with cluster-inducing properties, in case the authors find it useful to discuss [6]. ([6] is not SSL, but it may be of interest, according to your judgement).

[5] Kamnitsas et al, Semi-Supervised Learning via Compact Latent Space Clustering, ICML 2018.

[6] Kenyon-Dean et al, Clustering-Oriented Representation Learning with Attractive-Repulsive Loss, AAAI 2019 workshop.

Finally, I below provide a reference to a recent work on SSL in time-series, where the references therein can also help identify important related work [7].

[7]  Jawed et al, Self-supervised Learning for Semi-supervised Time Series Classification, PAKDD 2020.


Con.2:
Evaluation shows that no method is consistently better than the baseline AE. Which diminishes a lot the value in the technical contribution (e.g. the interest in the 2 novel losses). Even more, central claim of the paper about performance of the methods does not hold.

Specificially, the claim from the abstract “our methods can consistently improve k-Means clustering” is not True. In Fig 2 and 4 from evaluation, we clearly see that no method “consistently” improves over baseline AE across all settings. For example, DB index is clearly worse than all on ECH5000. Prototype loss also falls short, especially in UWave and Fig 4 ECG500. Silhouete loss also falls below AE on UWave in the little-labels regime in Fig2.a and across all settings in Fig.4.b. This actually shows that no model “consistently” improves AE.
Similarly, these claims about performance are accompanied in Abstract & Sec 1 with comments about the “maximum increase” in performance. But the maximum increase does not really convey the actual picture, as these are only the extrema. Instead, the authors should try to derive some statistic that represents the overall performance better.

Con.3:
Evaluation is limited:

Method is only against unsupervised CAE and Ren 2018. This is a result of limited literature review I guess (weak point 1). The authors could expand the evaluation by comparing to:
* other “shallow” approaches that are already applied in time-series. The authors mention some of these in the related literature, arguing they are slow. The authors could at least try some of them, and report performance and time-for-running. This is very important, as the time-to-run of these approaches should be compared with the time-to-train of the proposed approach, which is never mentioned. (to my understanding the authors train and evaluate on the same samples, ala transductive SSL, hence training time plays the role of time-to-run of the traditional approaches)
* other related losses for SSL in deep models (e.g. [5], or other similar easy to apply ones like VAT, see references in [5]), or other baseline generative models (e.g. VAE/GAN variants).

Con.4:
The hyperparameter study in Sec 4.2.2 is not really useful. The authors explore whether different number of layers in the AE affect performance, and whether updating the weights of the AE with the gradients of the SSL loss at every batch step or only every epoch makes a difference. This is not insightful about the proposed losses themselves.

Con.5:
There is currently no mention of an attempt to configure the model and training configuration for each loss best. E.g., learning rate, number of steps till convergence, etc. So, I am guessing, same learning rate etc is used in all settings, which is unclear at which settings they have been configured. But, I think that the losses may have gradients at different scale (I m not sure, but seems so). For example, L_proto in Eq.3 has the form ylogp and may have gradients that are at a different scale than Eq 5 and Eq 6 that directly take the gradient of the distances. This means that for optimal convergence, and hence fair comparison, gradients of each loss should be scaled differently. E.g. using different learning rate. Or, better, using a different weight multiplier in front of L_supp, so that the same learning rate can be used for the reconstruction loss across all settings and only scale the L_supp gradients independently. What I would normally expect is a study of the performance’s sensitivity on such as hyperparameter.

Con.6:
Reproducibility is not high. There is no code, and various parameters for training (learning rates, number of sgd steps, etc) are not included.

Con.7:
Clarity of the paper, especially Section 3 (method) is not good. It is largely due to not good mathematical notation, which could help define certain things well, coupled with a problem in Fig 1:

Sec 3.1 describes the CAE architecture, which is a very small model (2 layers of conv+pool) but still I don’t think it does it adequately. I am particularly confused about the description of the 1D conv filters, their width, the dimension along which the conv is applied etc. This description is done in natural language and I think it’s currently ambiguous. The authors could introduce some more formal definitions of the input tensor (X \in ??), weight tensors, the output tensor of a convolutions, define dimensionality of these tensors, which would clarify better the model’s operation.

Math notation should be improved. E.g. in Eq.1, C_j is not formally defined. Eq.2, p_j undefined (I think it’s the prototype for class j, where in Eq.1 is C_j, while p has been previously defined as a probality).

Eq 3, L_proto, is a semi-supervised loss (L + U).

Fig 1, shows only a loss that is called “semi-supervised” and symbolised as L_sup, while the figure shows that this loss is only dependent on embeddings of labelled data, not those of the unlabelled data. Hence it confuses, as it looks like a supervised loss.

j is sometimes a class (Sec. 3.2.1), sometimes a sample (Eq4), and vice versa for C_j. Also, C_j in 4.b. represents a set of samples, and not just a class number, as in Eq.1. Please, repass math notation and define carefully each term. The current state makes for a difficult reading experience.

\bar{C} in Eq.6 undefined.

##############################################
Questions for rebuttal period:

Please address the weak points I raised above, as well as the following (some are related to above anyway):

Can you please clarify whether all the time-series within a database (e.g. FacesUCR) are of the same length? I guess so, but I would like a clarification. Also, please clarify this in text, e.g. in 4.1.3, as it is important for the reader to understand whether all samples have same dimension.

Can you state explicitly in the text whether you are using any weighting of the SSL loss VS the reconstruction loss? I think not, but please state it explicitly. If not, please add a short discussion of whether you think the gradients of the different losses are at comparable scale, and whether such weight is needed or not.

Can you explain how were configuration parameters for the methods been performed? Since there is no weighting for each loss, I am particularly interested in things like the learning rate. Was it configured on the CAE? On the CAE+SSL loss? On each one separately? On which set, since there is no validation fold? Please add to text.

Sec. 4.2.2 “ Since autoencoder updates… for comparison”: Can you please clarify/rephrase what this means? Why at each epoch only? Do you mean specifically due to the SSL loss? Because in previous sentences you say that the autoencoder *still* gets updated at each batch via the reconstruction loss.

#############################################

Minors, or additional feedback for improving the work in the future (not subject to rebuttal):

Sec. 4.2.1. “This also contributes to the poor KNN performance on this dataset.”. I have not noticed any mention of KNN here (at least not until later in Fig.4). Where is this comment based on? If it refers to experiments later on (Fig 4), please clarify/refer to them in text.

Sec. 4.2.2. “The findings here also support … Prototype loss model.”: I think these results do not “support” previous results, as the experiments are made on the database the same database that showed the trend in the first place. If they were made on ECG5000, they would “support” another trend. I would remove this argument. What these results support is that the filter-size is not important on this database.

The authors could try to derive metrics that summarize performance across multiple settings. (e.g. “average” performance across all settings in each database, or average performance across databases on a particular setting (e.g. 4 labels)). Summarized in a table? It could help define clearly which method does better.

It would be interesting to explore how the amount of unlabelled data affects results of training.

It would be interesting to have 2 sets of unlabelled data, one for training and one for evaluation. To test whether the embedding generalizes and provides good clustering on unseen data, not just the ones it was trained on, and how the 2 cases (seen unlabelled vs on unseen unlabelled) compare.

Sec. 4.2.1: The authors provide a description of how training evolves (“During training, the clusters… ground-truth data”) in this case of sparse labels. Perhaps in future work it would be interesting to actually provide an empirical analysis of this behaviour (e.g. showing how clustering changes between epochs? Perhaps even on a toy-dataset, in a low-dimensional embedding, to observe how the losses behave).

It would be nice to have statistical significance tests between model results in Fig 2, especially about the main claims. (E.g. to prove that Silhouete is significantly better than others)

Typos and other minors:

Sec.1: “and is not always segmented”: Unclear to me what this means, perhaps rephrase a bit.

Some typos:

Sec. 1 “have been show” => shown

Sec.1 “an semi-supervised” => a

Sec. 1 “model semi-supervised model”

Sec. 1 “a unsupervised” => an

Sec 3.2.2 “the a partitioning”


**=========== Update to Review, after Updates to Manuscript during Rebuttal period ==========**

Summary of improvements during rebuttal and remaining concerns:

- Extended the literature review, with references to an unsupervised clustering method (k-shapes),  and methods that use CAE, LSTM and VAE. This is a significant improvement. The literature review still does not discuss previous losses that lead to clustering within the latent space, which is what the proposed methods here perform, and are hence closely related (see my initial comment).

- Rephrased/corrected claims about performance of the method. The corresponding claims about performance is now that the methods “can usually improve k-Means clustering performance”, improving the baseline CAE. This is now accurate. Yet, I do not think the performance is a sufficiently strong argument unfortunately (especially given the limited evaluation, only against few and basic baselines).

- Evaluation has been expanded by employing unsupervised clustering via k-Means, k-Means+PCA, and k-Shape in Table 1. However, there is still no comparison with any other more advanced method, such as SSL based, DL based, etc (as per my initial comment). I think there is a lot of improvement here, before the paper can display improvement over the current state of the art.

- No improvement on the ablation study, which is currently not particularly useful, e.g. by investigating aspects of the losses and their behaviour to provide more insights into the method, or performing empirical sensitivity-analysis to meta-parameter values.

- Added clarifications about some values hyper-parameters used (Adam optim with default TF2.1 learning rate for all). There seems there was no attempt to find optimal hyper-parameter configuration for each method, in order to perform fair comparisons (e.g. each method may require different weighting/learningrate). The authors seem to acknowledge the issue and defer it to future work. However, I think that without such investigation (e.g. via an empirical study and sensitivity to these hyper-parameters), we cannot be strongly confident about conclusions on the relative performance of methods, as they can be sensitive to configuration.

- Provided the code of the work as supplementary material, hence reproducibility is greatly increased. Thank you.

- Improved clarity of the paper by improving Fig 1 and some of the math notation problems.


Overall, the work has been improved during the revision, hence I will raise my rating (from 3 to 4). However, I believe the remaining problems of limited literature review, evaluation, no sensitivity analysis to hyper-parameters (or effort thereafter to configure them for each method) or other sort of empirical analysis that would give insights to the behaviour of the losses, still keep my score relatively low.

---

> ### Author Response · Authors · 2020-11-25
> **Thank you for your time, and for your comprehensive review and constructive criticism. We are much obliged.**
>
> * “Con.1: The work is not well positioned within the literature.”
>     * Thank you for your comprehensive review of these deep architectures. We have extended the Related Work Section accordingly. In particular, we add some discussion of VAE models to Section 2,  but as you mentioned our focus for this paper was primarily on how to improve the clustering ability of the CAE architecture. We have also added a discussion for examples of other uses of autoencoder architectures on time series data.
> * “Con.2: Evaluation shows that no method is consistently better than the baseline AE.”
>     * You are absolutely right. Our claim was not accurately worded in the original version of the paper. We have modified the wording of our claim to reflect the concerns about “consistently improving K-means performance”. Also we noted that the tables in the Appendix show that Silhouette Loss does on average always outperform the CAE, with the exception of the 4-examples case on UWave data.
> * “Con.3: Evaluation is limited:”
>     * We agree that the experimental evaluation could be more extended. Accordingly, we have performed new experiments to compare our proposed work versus related unsupervised approaches for time series clustering. The results of these experiments are added as a new table (Table 1) which shows the performance of some existing unsupervised time series clustering approaches versus our proposed solution. We have also added some discussion of these results to the paper in Section 4.2. Given that the focus of our paper is not on autoencoding part of the proposed model, we did not feel we need to compare with a VAE based solution.
> On the other hand regarding the existing semi-supervised solution, in the cases of the two methods discussed in the related work, we felt that Dau et al. (2016) was not scalable to large datasets or longer time series because of its reliance on DTW. Also, for He et al. (2019), the problem formulation is slightly different than ours, since the semi-supervised elements are implemented as constraints (must-link and cannot-link) between pairs of samples, rather than a set of labels. Therefore, these approaches do not seem to be comparable with our proposed work. Instead we felt that the Prototype Loss was the most comparable approach with our proposed solution, and hence, we used it as a reference for comparison in our experiments which uses traditional class labels.
>
>
> * “Con.4: The hyperparameter study in Sec 4.2.2 is not really useful.”
>     * We agree that the selected hyperparameters might not be most useful for all readers. Nevertheless we felt some readers might benefit from these experimental results.
> * “Con.5: There is currently no mention of an attempt to configure the model and training configuration...”
>     * Thank you very much for pointing out these important issues with lack of characterization of our experiments. We have updated the text in Section 4.1.1 to give more details about model training as follows: “For optimization, we use the Adam optimizer as implemented in Tensorflow 2.1.0. We use the default learning rate of lr=0.001 for all experiments, and train for 200 epochs...”. We also agree that it is important to explore the weighting of the losses. However, we would like to defer this extension of the study to future work. We have included this in the list of items we have referenced in the “Conclusion” Section of the revised paper. In particular we will explore weighting the losses and compare the scale of the gradients then.
> * “Con.6: Reproducibility is not high. There is no code, and various parameters for training (learning rates, number of sgd steps, etc) are not included.”
>     * Thank you for your note. As mentioned above, we have updated the text in Section 4.1.1 to give details about model training. Also we have made our code available as supplementary material, along with the three datasets from UCR Archive.
> * “Con.7: Clarity of the paper, especially Section 3 (method) is not good”
>     * Thank you for your note. We feel Figure 1 in the original version of the paper has caused confusion. We have updated Figure 1 as well as some of the referenced mathematical notations for clarity accordingly.

---

> > ### Author Response · Authors · 2020-11-25
> > **To address the specific questions raised at the end of the review:**
> >
> > 1. Yes, all time series within a dataset are the same length. This is a requirement of the CAE architecture. However, we note that datasets with varying lengths may still be used, but some pre-processing step would be required to normalize the lengths of the samples first.
> > 2. We do not use any weighting of the losses. We found that in practice the model did not have issues with either loss dominating the gradients. However, as mentioned above we are planning to explore this in extensive details using experimentation.
> > 3. We used the Adam optimizer as implemented in Tensorflow 2.1 for our experiments. All experiments use the default learning rate of 0.001. We add text to clarify this at Section 4.1.1
> > 4. Thanks for your comment. This was an error on our part. The text has been updated to reflect that the autoencoder updates at the end of each batch. The experiment is testing the gradient update strategy for the semi-supervised losses only, as the autoencoder gradient is calculated and parameters updated at each batch. In the experiment where no semi-supervised loss is in use, the performance between the two update strategies will be identical.

---

### Official Review · AnonReviewer4 · 2020-10-28
**Interesting work towards clustering time-series data**

**Rating:** 6
**Confidence:** 3

**Review:**

Summary:

This paper proposes a semi-supervised architecture for clustering time-series data based on Convolutional Autoencoders (AEs). The model combines the regular reconstruction loss usually employed in AEs with two new losses based on the intrinsic clustering evaluation metrics, Silhouette and DBIndex. The experiments show that this setup can achieve good clustering results (ARI) even when very few labeled examples of each class (4-28) are provided.

Positive points:

Overall, the presentation of the paper is good, it is well organized and easy to understand.
The experiments are plenty and well thought to evaluate the proposed model in three different datasets.

Questions:

In Equation 2b, shouldn’t be C_j and C'_j instead of p_j and p'_j? Otherwise, what p_j means?
Is the number of clusters defined by the number of classes? What happens if a class is originally split into multiple clusters?

Points to improve:

In “Both LSTM and convolutional autoencoders have been show to be successful at learning latent representations of time series data.” Please, provide at least one reference for each type of model.

Evaluating clustering quality is a tricky task. Using only ARI might be misleading, as ARI and most clustering metrics, has its problems. Therefore, I would suggest adding other metrics such as Normalized Mutual Information (NMI), Purity, and Clustering Accuracy.

A comparison with previous methods is lacking. Although I’m not aware of other methods for semi-supervised clustering of time-series data. There several options to compare with in the unsupervised clustering case. The closes one might be Fortuin et. al. (2019) “SOM-VAE: Interpretable Discrete Representation Learning on Time Series”.

A few typos and bad formating to fix:
“have been show“-> "have been show" (section 1)
“an semi” -> “a semi” (section 1)
“squnces” -> “sequences”
fix reversed quotes as the first quote in “reversed" (section 3.1)
“equation (5)” -> “Equation (5)” (section 3.3.2)

---

> ### Author Response · Authors · 2020-11-25
> **Thank you very much for your valuable review and for your time. To address your concerns:**
>
> * “In Equation 2b, shouldn’t be C_j and C'_j instead of p_j and p'_j?”
>     * Yes, we have corrected the error in Equation 2b. The equation should use the prototypes C_j instead of the probabilities p_j. The number of clusters used in the optimization is identical to the number of classes in the dataset. We assume that all labeled samples will have a single label, and that labels are mutually exclusive (i.e. a sample does not belong to more than one class).
>
> * “In “Both LSTM and convolutional autoencoders have been shown to be successful at learning latent representations of time series data.” Please, provide at least one reference for each type of model.”
>     * Thanks for your comment. We have added references for well-known prior work utilizing these architectures in Section 2.
> * “Evaluating clustering quality is a tricky task. Using only ARI might be misleading...”
>     * Thank you for your comment. In earlier experiments, we evaluated both NMI and ARI, and found that the values were similar for both across a variety of datasets, so we only reported ARI for our experiments.
> * “A comparison with previous methods is lacking. Although I’m not aware...”
>     * Thank you for your comment. We have extensively revised the Related Work Section of the paper to discuss other existing methods and compare them with our proposed work. We have also performed new experiments to compare our proposed solutions with existing solutions (please see Table 1 and Section 4.2 for more details).
> * “A few typos and bad formatting...”
>     * Thank you, we have corrected these issues.

---

### Official Review · AnonReviewer1 · 2020-10-28
**Benchmarking different clustering losses in semi-supervised time series data clustering**

**Rating:** 5
**Confidence:** 2

**Review:**

This paper benchmarked three different existing losses, DB/Propotype/Sihouette, on time series clustering. Although there is no technique innovation, this is an interesting topic and the results look good to me. However, there are some concerns:

1. I am not an expert in time series clustering, especially in the semi-supervised clustering case. As there is a labeled dataset, the reason to solve this problem as two steps is not clear to me: 1. learning the semi-supervised representation and 2. do the clustering. Why not just benchmarking by semi-supervised learning metric, like accuracy?

2. The effect of the clustering method looks unclear. In other words, will the conclusion be held on different clustering methods, rather than K-means.

3. Different losses have different performance rankings in different numbers of labels and datasets (Figure 2). Picking up the best and claiming the advantage does not form a fair comparison. It will be more interesting if the author can propose a method, which can consistently win other methods.

Minor concerns:

1. the bar plot should have space between each category.
2. it will be more convincing if baselines from other papers are considered.

---

> ### Author Response · Authors · 2020-11-25
> **Thank you very much for your valuable review. To address your concerns:**
>
> * “As there is a labeled dataset, the reason to solve this problem as two steps is not clear to me...”
>     * We would like to explain that although the datasets we used for evaluation have labels for all samples, the intended real-world use of our model is to cluster data where not all of the samples are labeled. In this use case, a domain-expert could choose a small subset of the samples to provide labels for, and the rest of the samples would remain unlabeled. We completely labeled datasets for our evaluation so that we can measure how the performance of the model changes with different amounts of labeled data. Since this may have not been explained properly in the original version of the paper, we have added some clarifying text to the beginning of Section 4.1.3 elaborating on the same: “All UCR Archive datasets are labeled, which is useful for our evaluation since we may experiment with differing amounts of labeled data. In a real-world scenario with unlabeled data, domain experts provide label information for a small subset of the data.”
> * “The effect of the clustering method looks unclear. In other words, will the conclusion be held on different clustering methods, rather than K-means.”
>     * Yes, the choice for the final clustering algorithm applied is left up to the user. Any existing clustering algorithm may be applied to the trained latent space, and the clusters obtained should be similar to those obtained through K-Means, since the latent space is optimized on its ability to partition the data. We noticed that this clarification was not included in the original version of the paper; thanks for your note. We have added clarifying text to Section 4.1.2: “We use the K-Means algorithm because the centroid-based nature of K-Means is a natural fit for the proposed losses. Notably, the Prototype Loss corresponds almost exactly to a K-Means objective, and both Silhouette and DB Index loss also rely on notions of cluster density around a centroid. However, any other general clustering method may be applied.”
> * “Different losses have different performance rankings in different numbers of labels and datasets...”
>     * We agree that it would be ideal for us to find a universal solution. However, as we show in our experiments, it seems different datasets require different loss function solutions with different logic. That’s why we feel both proposed loss functions are applicable and dominating under relevant circumstances dictated by the data requirements. In particular, in the case of our two proposed methods, we find the Silhouette loss works best generally. Silhouette loss optimizes a per-sample measure cluster label and data agreement, while DB Index loss optimizes separability between pairs of clusters. We think that DB Index may perform better in cases with smaller numbers of clusters, while Silhouette may work better with larger numbers of clusters.
>
> * “Minor concerns”
>     * Thank you for your note. We have addressed your concerns. In particular, we have performed new experiments to compare our proposed solutions with existing solutions that can serve as baseline solutions (please see Table 1 and Section 4.2 for more details).

---

### Official Review · AnonReviewer3 · 2020-10-29
**A simple semi-supervised framework for time-series clustering. Evaluation is not adequate to get conclusive results of advancement of the state of the art in any way**

**Rating:** 4
**Confidence:** 5

**Review:**

The paper presents an autoencoder-based approach for semi-supervised time-series clustering. Specifically, the paper exploits convolutional autoencoder (CAE) and integrates internal validity indexes to evaluate clustering quality in addition to small number of labeled data. Results on three datasets show improvement over the unsupervised case.

Pros

- Well-written paper that tackles an important problem
- Study of validity clustering indexes and their integration to current CAE architectures
- Results on three datasets showing the potential of this methodology

Cons

- Missing related work
- No comparison against unsupervised time-series clustering methods
- No comparison against semi-supervised time-series clustering methods
- Only three datasets used despite 100+ available

Details:

- The paper is missing a decade of progress in the area of unsupervised time-series clustering.

[a] Paparrizos, John, and Luis Gravano. "Fast and accurate time-series clustering." ACM Transactions on Database Systems (TODS) 42.2 (2017): 1-49.

- Comparison against unsupervised methods such as k-means and k-shape [a] is necessary to understand the effectiveness of CAE at the first place.

- Comparison against semi-supervised methods reviewed in related work is needed.

- Evaluation on all 100+ datasets of the UCR archive is necessary. To advance the state of the art, the method needs to show improvement in the entire benchmark and not just on three datasets, which we are not sure how exactly they were picked and why.

---

> ### Author Response · Authors · 2020-11-25
> **Thank you very much for your time and valuable review.**
>
> To address your concerns:
> * “Missing related work”
>     * We have extensively revised the Related Work Section to include the missing related work, including the recommended citation.
> * “No comparison against unsupervised time-series clustering models”
>     * We have performed new experiments to compare our proposed work versus related unsupervised approaches for time series clustering. The results of these experiments are added as a new table (Table 1) which shows the performance of both K-Shape and K-Means on the raw data. We have also discussed these results in Section 4.2.
> * “Comparison against semi-supervised methods reviewed in related work is needed.”
>     * In the cases of the two methods discussed in the related work, we felt that Dau et al. (2016) was not scalable to large datasets or longer time series because of its reliance on DTW. For He et al. (2019), the problem formulation is slightly different than ours, since the semi-supervised elements are implemented as constraints (must-link and cannot-link) between pairs of samples, rather than a set of labels. We felt that the Prototype Loss was the most comparable approach with our proposed solution, and used it as a reference for comparison in our experiments.
> * “Evaluation on all 100+ datasets of the UCR archive is necessary. To advance the state of the art, the method needs to show improvement in the entire benchmark and not just on three datasets, which we are not sure how exactly they were picked and why.”
>     * We chose the three datasets because they have a relatively large number of samples compared to the other datasets in the UCR Archive. Other datasets in the archive would not lend themselves well as test datasets to challenge our proposed solution, therefore, not included. We have added clarifying text to Section 4.1.3 explaining the same. “In the case of trainable architecture like our proposed model, large datasets are advantageous, as larger numbers of samples will increase the quality of the latent featurization, and help to improve generalization of the features for unseen samples.”

---

### Decision · Program_Chairs · 2021-01-07
**Final Decision**

**Decision:**

Reject

**Comment:**

This paper addresses an important problem of semi-supervised learning of time-series data.  Their approach is based on a convolution autoencoder for learning a time-series latent space.  To guide learning an appropriate embedding, they explore three alternative internal clustering metrics (prototype loss, Silhouette loss, and DB index loss) coupled with the autoencoder reconstruction loss.

The approach is reasonable and interesting, however as pointed out by the reviewers, the current submitted version needs major revision for it to be accepted.  Key weaknesses are:
1.	It lacks an extensive literature review.  The reviewers have made several suggestions for improving this.
2.	The experiments are weak.  First, state-of-the-art baselines are missing.  Second, additional alternative evaluation metrics will strengthen the evaluation of unsupervised methods (e.g., adding NMI, accuracy, cross-validation of internal metrics).  Providing evaluation of when which loss would work better on what types of data would provide insight to the various losses proposed.